# Thyme Essential Oil Reduces Disease Severity and Induces Resistance against *Alternaria linariae* in Tomato Plants

**Luis Alberto Saltos-Rezabala** [1] , **Patrícia Ricardino Da Silveira** [1] , **Dérica Gonçalves Tavares** [2] , **Silvino Intra Moreira** [1] , **Thiago Alves Magalhães** [2] , **Deila Magna Dos Santos Botelho** [1] and **Eduardo Alves** [1,*]

1    Departamento de Fitopatologia, Universidade Federal de Lavras—UFLA, Lavras 37200-900, MG, Brazil
2    Departamento de Biologia, Universidade Federal de Lavras—UFLA, Lavras 37200-900, MG, Brazil
*    Correspondence: ealves@ufla.br

**Abstract:** Currently, the use of alternative tools for chemical control has become one of the most sustainable and eco-friendly options for crop protection against phytopathogens. Thus, the present study aimed to assess the effect of essential oils (EOs) on the control of tomato early blight (EB), a highly destructive disease of this crop. The antifungal activity and ability to induce resistance induction of the EOs of thyme (*Thymus vulgaris*), lemongrass (*Cymbopogon citratus*) and tea tree (*Melaleuca alternifolia*) were tested for control of *A. linariae* in tomato plants. In vitro, mycelial growth and conidial germination were strongly inhibited when thyme EO (TEO; 2000 ppm) and lemongrass (LGEO; 2000 ppm) were applied. The infected leaf area and final disease index were decreased by 47.78% and 39.00%, respectively, compared to the water treatment. Foliar spraying with TEO increased the activity of the defense-related enzymes peroxidase, polyphenol oxidase and β-1,3-glucanase. Furthermore, callose deposition and phenolic compound accumulation in tissues infected by *A. linariae* improved after TEO application. In conclusion, TEO stimulated the defense system of tomato plants infected by *A. linariae*, which was associated with a reduced severity of EB. These results indicate that TEO is a potential tool in tomato EB disease management.

**Keywords:** *Thymus vulgaris*; antifungal activity; resistance induction; early blight; alternative control

## 1. Introduction

The tomato (*Solanum lycopersicum* L.) is the world's most economically important vegetable [1]. World tomato production in 2020 was 186,821,216 tons, with China and India as the largest producers [2]. Tomatoes provide a series of benefits to human health through the provision of antioxidant substances such as lycopene, β-carotene and phytoene [3]. Tomato production is affected by numerous fungal diseases such as tomato early blight (EB), caused by *Alternaria* spp. [4], manly *Alternaria solani*, being one of the most common and which has caused a reduction in the yield and quality of agricultural Brazil production. *A. solani*, as it is widely known, was reclassified by Woudenberg et al. [5] due to the size of its spores, while the *A. lineariae* species mainly occurs in tomato crops in Brazil. EB is common in environments with high relative humidity, rainfall and high temperatures [6]. In some cases, annual losses of economic income due to EB have been estimated to be up to 79% [7]. The characteristic symptoms of EB on leaves are brown to black spots with concentric rings and a slight depression in the center of the lesion, and in advanced stages of the disease, severe defoliation can occur [8].

Currently, EB management is based on the use of cultural practices and chemical control [7]. However, the evolution of fungicide resistance in phytopathogens is a major concern in sustainable disease management [9]. Chemical control has resulted in the emergence of resistance to fungicides due to selection pressure, e.g., strobilurins, boscalide and triazoles [10–12]. Additionally, some synthetic chemicals have been reported to be toxic to beneficial and nontarget organisms such as natural enemies and pollinators [13]. Modern

crop protection focuses on sustainability and the use of eco-friendly pesticides, mainly the application of nonchemical methods of plant protection against pests and diseases [14]. Although biological control through biological control agents (e.g., *Bacillus* spp.) has shown high efficiency in the management of plant diseases with low environmental impact, it is already a well-established and well-studied control method [15–19]. In addition to biological control, new products have emerged as alternative and promising molecules for phytopathogen control, including essential oils (EOs) from plants [20].

EOs are mixtures of volatile organic substances produced by the secondary metabolism of some plant species, particularly aromatic plants [21,22]. The abundant bioactive components of EOs have been documented for their antimicrobial action against a broad spectrum of pathogens [13,23]. The antifungal properties of EOs have been shown to control phytopathogenic fungi such as *Alternaria* spp. [13,24–28], *Fusarium* spp. [29–31], *Colletotrichum* spp. [32], and *Aspergillus* sp. [33] under in vitro and in vivo conditions in different cultivated plants.

EOs have several components with multiple mechanisms of action, that affect different cellular targets and disrupt fungal biological functions and processes [34]. EOs can increase the permeability and denaturation of cell membrane proteins, affect energy metabolism by altering electron transfer and ATP synthesis, damage genetic material by inhibiting gene expression, and cause cell wall damage [35]. For example, thymol is a monoterpene that can bind to hydrophobic regions of proteins and dissolve in lipid phases [36]. The activity of oxygenated monoterpenes affects cell wall synthesis, blocks enzymatic activity, and leads to a rearrangement of lipids, which, as a consequence, changes cell membrane properties and functions [27].

On the other hand, the indirect action of bioactive components of EOs can stimulate cellular defense responses against pathogen attack [14]. For example, thyme EO (TEO) treatment increased peroxidase (POX) activity which resulted in reduced severity of gray mold and Fusarium wilt caused by *Botrytis cinerea* and *Fusarium oxysporum* f. sp. *radicis lycopersici*, respectively [37]. *Cinnamomun zeylanicum* EO increased the activity of catalase, ascorbate peroxidase, phenylalanine ammonia lyase and POX in tangerine plants for *A. alternata* control [38].

Products of organic origin such as EOs are increasingly gaining interest in the search for alternatives to synthetic chemical pesticides, considering their multiple modes of action against phytopathogens [14]. Therefore, the use of EOs could become an environmentally sustainable method for crop protection and, consequently, reduce the risk to human health [21]. Despite significant advances in the control of phytopathogens in the last decade, particularly under in vitro conditions, in vivo or in planta efficacy tests are still limited [22]. Thus, the present study aimed to assess the antifungal effects of EOs on *A. linariae* control and the biochemical responses in tomato plants.

## 2. Materials and Methods

### 2.1. Essential Oils, Biological Control Agent and Fungicides

The essential oils (EOs) of thyme (*Thymus vulgaris*), lemongrass (*Cymbopogon citratus*) and tea tree (*Melaleuca alternifolia*) were acquired from companies registered in the local trade of Brazil (Table 1). The biological control agent (BCA) *Bacillus amyloliquefaciens* MBI600 (Duravel®), and the synthetic fungicides tebuconazole (Folicur®) and azoxystrobin (Amistar®), were used as control, biological treatment and chemical treatment products, respectively.

**Table 1.** Composition of essential oils, biological control agent and commercial chemical fungicides.

| Eos *, BCA † and Fungicides ‡ | Active Compounds | Manufacturer |
|---|---|---|
| Thyme (*Thymus vulgaris*) * | p-cymene 37–40%, 1,8-ceneole 1–2%, camphene 1%, myrcene 1–2%, linalool 3–4%, thymol 45–48%, limonene < 1%, α and β-pinene 4–5%. | LASZLO |
| Lemon-grass (*Cymbopogon citratus*) * | Myrcene 14–20%, citronellal < 3%, neral 26–32%, geranial 32–38%, linalool 1–2%, pinene 3–5%, geraniol 2–3%, borneol < 3%. | LASZLO |
| Tee tree (*Melaleuca alternifolia*) * Timorex gold® | *M. alternifolia* oil extract 222.5 g/L (22.25%). | STK bio-ag Technologies |
| *Bacillus amyloliquefaciens* (Duravel®) † | *B. amyloliquefaciens* MBI600 ($5.5 \times 10^{10}$/g) - 110 g/Kg (11%). | BASF |
| Tebuconazole (Folicur®) ‡ | Tebuconazole 200 g/L (20%). | Bayer |
| Azoxystrobin (Amistar®) ‡ | Azoxystrobin 500 g/kg (50.0%). | Syngenta |

* (EO) Essential oil; † (BCA) biological control agent; ‡ commercial synthetic fungicide.

### 2.2. Fungal Isolate, DNA Extraction, Amplification, and Phylogenetic Analyses

An *Alternaria* sp. isolate with high aggressive potential was supplied by Embrapa Hortaliças (Brasília, Distrito Federal, Brazil). The fungus was deposited in Coleção Micológica de Lavras (CML) at the Departamento de Fitopatologia, Universidade Federal de Lavras, MG, Brazil and registered with the accession number CML 4259. Genomic DNA was obtained using the procedure of Leslie and Summerell [39]. The genes for the Alternaria major allergen (Alt a1) and glyceraldehyde-3-phosphate dehydrogenase (gpd) were amplified using the primers Alt-for/Alt-rev and gpd1/gpd2 according to Hong et al. [40] and Berbee et al. [41], respectively. The actin (ACT), plasma membrane ATPase (ATPase) and calmodulin (CAL) genes were amplified using primers designed by Lawrence et al. [42]. The polymerase chain reaction (PCR) conditions were: initial denaturation at 95 °C for 1 min, 35 cycles at 95 °C for 30 s, 55 °C for 30 s, and 72 °C for 1 min, and a final extension at 72 °C for 2 min. PCR was performed using Taq Pol-Hot Start Master Mix (Cellco), and each PCR contained 5 μL of reaction buffer, 1 μL of each primer (10 μM), 1 μL of dNTPs (200 mM), 1 μL of DNA template (20 ng/μL), and nuclease-free water to obtain a final volume of 50 μL. PCRs were performed using the Applied Biosystems Veriti 96-Well Thermal Cycler (Fisher Scientific AS, Oslo, Norway). Amplified fragments were purified using the Wizard® SV Gel and PCR Clean-Up System (Promega, Madison, WI, USA), and sequenced by ACTGene Co. Ltd. (Alvorada, RS, Brazil). Forward and reverse sequences of each region were assembled using SeqAssem ver. 07/2008 (SequentiX, Digital DNA Processing, Klein Raden, Germany) to obtain the consensus sequences. The sequences generated in this study for Alt a1, *gpd*, ACT, ATPase, and CAL were deposited in GenBank, and additional sequences for phylogenetic analyses were also obtained from the GenBank database (Supplementary Table S1). Sequences were aligned using CLUSTAL W (implemented in MEGA X) and phylogenetic analyses were performed using MEGA X software [43]. The datasets consisted of ACT (12 parsimony informative positions/939 bp), Alt a1 (13/472), CAL (25/764), gpd (12/576), and ATPase (29/1204). Phylogenetic trees were constructed based on maximum parsimony (MP) analysis for each gene and for the concatenated sequence. Clade support was based on 1000 bootstrap replications.

### 2.3. Inoculum Preservation and Preparation

The fungal isolate was preserved using the Castellani method [44]. For conidia production, the fungus was grown for 15 days in oat agar culture medium (40 g of oat flakes, 1 L of distilled water, and 16 g of agar) under continuous ultraviolet light at a temperature of 25 °C.

### 2.4. In Vitro Evaluation of the Antifungal Activity of Essential Oils against the Fungal Isolate

In the mycelial growth inhibition (MGI) assay, concentrations of 0, 500, 1000, 2000 and 4000 ppm were tested for all EOs. For the control treatments, BCA MBI600, tebuconazole and azoxystrobin were added at concentrations of 1500, 1000 and 120 ppm, respectively, following the manufacturer's recommendations. The EOs were previously diluted in Tween 80 (0.01%) before being added to the potato dextrose agar (PDA) culture medium [29]. One disc (5 mm Ø) of culture medium with fungal growth was placed in the center of the Petri dish (90 mm Ø) with BDA culture medium. Petri dishes were sealed with parafilm and placed in a BOD incubator at 25 °C ± 1 °C. Eight days after inoculation, the diameter of the colonies was measured with a digital caliper. The MGI was estimated using the method described by Ji et al. [45] (Equation (1)).

$$MGI = (Ctrl - Trt)/Ctrl * 100 \tag{1}$$

where Ctrl is the control colony growth and Trt is the treatment colony growth. From the MGI values, the half maximal effective concentration ($EC_{50}$) was calculated using Graphpad Prism software (version 6.0, San Diego, CA, USA) (Equation (2)).

$$Y = 100/(1 + 10^{(X-LogEC50)}) \tag{2}$$

In the conidial germination assay, aliquots with the concentrations mentioned above were added to the potato dextrose broth (PDB) culture medium (KASVI®, Laboratorios Conda S.A, Barcelona, Spain). One milliliter of PDB culture medium was added to each well of cell culture plates (K12-024, Kasvi®). The same procedure was performed for the control treatments. Then, a 0.1 mL aliquot with a concentration of $10^5$ conidia/mL of the fungal isolate was added to each well of the plates where the respective treatments were located. Immediately, the plates were incubated in a TECNAL TE-420 incubator at 25 °C ± 2 °C and 110 rpm for 14 h without a photoperiod. After 14 h, germination was stopped by adding lactoglicerol. Slides for observation and conidia counting under the microscope were mounted (Carl Zeiss Primo Star, Jena, Germany). Conidia that presented a germ tube with a length greater than half the size of the conidia were considered germinated. A total of 150 conidia were counted, considering 50 conidia per replicate. The MGI and conidial germination experiments were performed in a completely randomized design (CRD). All experiments were repeated three times.

### 2.5. In Vivo Evaluation of the Antifungal Activity of Essential Oils against the Fungal Isolate
2.5.1. Experimental Conditions

The experiment was carried out during the spring and summer of 2021 in Lavras city, MG, Brazil, located at 21°14′43 south and 44°59′59 west and at 919 m altitude. An initial screening method was developed for the selection of treatments with greater control efficiency (data not shown). From this selection, treatments including in thyme EO (TEO), BCA MBI600 and tebuconazole were chosen for further experiments. Tomato seeds from var. Santa Clara were previously disinfected in ethanol (70%) and NaClO (1.5%) and rinsed three times in sterile distilled water.

The seeds were planted in 128-well polystyrene trays. Fifteen days after sowing, the seedlings were transferred to 5 L pots with vegetable substrate (Tropstrato HT; Vida Verde, Campinas, SP, Brazil), pH 5.8. The tomato plants were grown throughout the season in a greenhouse at 25 ± 4 °C, 70% relative humidity and a 12 h photoperiod. Plants were irrigated when necessary, based on water requirements at each growth stage. Fertilization at transplanting was performed with single superphosphate (18P-16Ca-10S) and 04-14-08 (NPK). From the beginning of flowering, the plants were fertilized with 20-0-20 and supplementary foliar fertilization with micronutrients performed at 15-day intervals. The treatments were distributed in a CRD with six replications. Each repetition was represented by one plant. The experiment was repeated once.

### 2.5.2. Plant Inoculation

The inoculum preparation was developed following the procedures of Marchi et al. [46]. Twenty days after transplantation, plants at the V5 phenological stage (BBCH scale) [47] were inoculated with conidia suspension at a concentration of $5 \times 10^3$ conidia/mL + Tween 80 (0.01%) using a hand sprayer. After inoculation, the plants were kept in a growth chamber for 24 h at 25 °C and 95% relative humidity with no photoperiod. After this period, the plants were returned to the greenhouse.

### 2.5.3. Application of Treatments

Twenty-four hours after inoculation, foliar applications with TEO, BCA MBI600 and tebuconazole were performed at concentrations of 2000 ppm, 1500 ppm and 1000 ppm, respectively, following a maximum protection spray program [48]. TEO and BCA MBI600 were applied weekly, while tebuconazole was applied at 14-day frequencies, following the manufacturer's recommendations. An adhesive spreader (Helper$^{®}$) was added to the application solution to reduce the surface tension of the water.

### 2.5.4. Disease Assessment

Seven days after inoculation, the infected leaf area was calculated using image analysis, according to the method described by Olivoto [49]. Additionally, disease severity was assessed weekly using a 0–5 rating scale, where 0 = no symptoms visible on the leaf; 1 = 1–10% of leaf area affected; 2 = 11–25% of leaf area affected; 3 = 26–50% of leaf area affected; 4 = 51–75% of leaf area affected; and 5 = > 75% of leaf area affected (Vakalounakis, 1983). The scores recorded in the disease severity assessments were converted to the disease index (ID) [50]. These indices were used to estimate the area under disease progress curve (AUDPC) [51] (Equations (3) and (4)).

$$ID = \Sigma \ (f * v)/(n * x) \times 100 \tag{3}$$

where f = number of leaves with a particular score; v = infection level (score); n = total number of leaves evaluated; and x = maximum infection level (score).

$$AUDPC = \Sigma \ (X_i + X_{i+1}/2) \ (t_{i+1} - t_i) \tag{4}$$

where $X_i$ is the severity score in the i-th observation; and $t_i$ is the time of i-th observation.

For the analysis of disease progression, the disease progression rate (r) was calculated using the exponential, logistic and Gompertz epidemiological models (Equations (5)–(7), respectively), according to Bowen [52].

$$y_t = y_0 \ e^{(rt)} \tag{5}$$

$$y_t = 1/\{1 + \exp[-(\ln(y_0/(1 - y_0)) + rt]\} \tag{6}$$

$$y_t = \exp[\ln(y_0) \ e^{-rt}] \tag{7}$$

where r is a rate parameter; $y_t$ is the disease proportion (0 < y < 1); $y_0$ is the initial disease level; t is the time; and e or exp indicate the exponential function, which is in natural logarithm (ln). The best model was chosen to represent the disease progress in relation to the studied treatments.

### 2.5.5. Evaluation of Vegetative Growth and Yield Parameters

Multiple growth and yield parameters were evaluated. Plant height (cm) and stem diameter (mm) parameters were registered at 50 days after inoculation (DAI). The number of leaves, fresh and dry weight of the plants (g), number of fruits, average weight of fruits (g), and productivity/plant (kg) were recorded at the end of the crop cycle (70 DAI).

### 2.6. Biochemical Analysis of Defense-Related Enzyme Activity

2.6.1. Obtaining Enzymatic Extracts

Samples of infected or uninfected plants were collected at 48, 96, 144 and 192 h after inoculation. Crude enzyme extracts were obtained by homogenizing 0.3 g of leaves in liquid $N_2$ in a mortar. Two milliliters of extraction solution, consisting of 0.1 mM EDTA, in 0.1 M potassium phosphate buffer, pH 6.8, 1 mM phenylmethylsulfonyl fluoride and polyvinylpyrrolidone (1%) was added. The homogenate was centrifuged for 20 min at $12,000\times g$ at 4 °C, and the supernatant was used for enzymatic evaluations and for protein dosages.

2.6.2. Determination of Enzyme Activity

*Peroxidases*: Peroxidase activity was determined following the procedure described by Kar and Mishra [53]. Aliquots of 2.5 μL of the crude leaf enzyme extract were added to 197.5 mL of a reaction mixture consisting of 70 mM potassium phosphate buffer pH 6.8 (137.5 mL), 50 mM guaiacol (30 mL) and 250 mM $H_2O_2$ (30 mL). The increase in absorbance at 480 nm and 30 °C was measured during the first minute of the reaction by purpurogallin production. Enzyme activity was calculated using a molar extinction coefficient of 2.47 $mM^{-1}$ $cm^{-1}$ [54] and the result was expressed as mmol $min^{-1}$ $mg^{-1}$ protein.

*Phenylalanine ammonia-lyase*: the activity of phenylalanine ammonia-lyase was determined according to El-Shora [55]. Aliquots of 9 μL of the enzyme extract were added to 161 μL of a mixture consisting of 40 mM Tris-HCl pH 8.8 (110 μL) and 87.5 mM L-phenylalanine (51 μL), with a final reaction volume of 170 μL. The reaction mixture was incubated at 37 °C for 60 min. The increase in absorbance was determined at a wavelength of 290 nm in the spectrophotometer, to verify the formation of trans-cinnamic acid. The molar extinction coefficient of 104 $mM^{-1}$ $cm^{-1}$ [56] was used for the calculations and the results were expressed in mmol $min^{-1}$ $mg^{-1}$ protein.

*Polyphenol oxidase*: the polyphenol oxidase activity was determined according to the method described by Kar and Mishra [53]. Aliquots of 20 μL of the enzyme extract were added to 180 mL of a reaction mixture consisting of 70 mM potassium phosphate buffer (140 μL) and 20 mM catechol (40 μL). The increase in absorbance was determined at a wavelength of 410 nm in the spectrophotometer over a period of 10 min at 30 °C. The molar extinction coefficient 2.47 $mM^{-1}$ $cm^{-1}$ [54] was used for the calculations and the results were expressed in mmol $min^{-1}$ $mg^{-1}$ protein.

β-1,3-glucanase: β-1,3-glucanase activity was determined using the method described by Lever [57]. The reaction was initiated by adding an aliquot of 5 μL of enzyme extract to a reaction mixture consisting of 57.5 μL of 50 mM sodium acetate buffer pH 5, 62.5 μL of laminarin (4 mg $mL^{-1}$) and 125 μL of 3,5-dinitrosalicylic acid (NDS). This reaction mixture was incubated at 45 °C for 60 min. Then, the mixture was incubated at 100 °C for an additional 15 min. The reaction was stopped by placing the samples in an ice bath. Control samples were prepared in the same way; however, the extract was added after heating the reaction mixture to 100 °C. The absorbance of the product released by GLU was measured at 540 nm in a spectrophotometer, and its activity was expressed in ΔAbs $min^{-1}$ $mg^{-1}$ protein.

### 2.7. Histochemical Analysis of Leaf Tissues Infected or Not by the Fungal Isolate

For the histochemical analysis, leaves were collected from plants infected or not infected with tomato early blight. Samples were collected at 48 and 192 h after inoculation. Leaf sections were fixed in Karnovsky's solution containing 2.5% glutaraldehyde, 2% paraformaldehyde and 0.05 M cacodylate buffer pH 7.2 + 0.001 M $CaCl_2$ for 72 h. After fixation, dehydration in an ethanol series (25, 50, 75, 90, and 100% 2x) and incorporation in methacrylate resin (Historesin; Leica) was performed.

In the infiltration process, 100% alcohol was replaced with 1:1 resin, and the sections were placed under vacuum for 5 min (3x) and kept in the refrigerator for 24 h. The resin

was replaced by pure activated resin, and the sections were placed under vacuum for 5 min (3x) every 10 min and kept in the refrigerator for 72 h.

For the blockage, the previously prepared and refrigerated inclusion resin was used. Subsequently, sections of plant material were placed inside each "shape" of the histomold, and covered with the blockage resin. The histomold was placed in an oven (50 °C) for 24 h. The blocks were removed from the histomold, adhered to wood with Super Bonder® glue and identified.

After drying, the blocks were sectioned in the microtome, obtaining 8 μm cuts. Then, these sections were mounted on microscope slides (25.4 × 76.2 mm). For the detection of phenolic compounds, ferric chloride reagent was used, following the method described by Marques and Soares [58]. Callose accumulation was determined by staining tissues with aniline blue (0.005%), following the method described by Eggert et al. [59]. Images of the plant tissue response to the fungus were obtained using an Epi-Fluorescence microscope Carl Zeiss MicroImaging Z1 (Carl Zeiss, Göttingen, Germany).

### 2.8. Data Analysis

The normality and homogeneity assumptions of variances were analyzed. The values of the repeated experiments over time were analyzed together after the residual variance homogeneity test and Hartly's Fmax test. Analysis of variance (ANOVA) was performed, followed by the Scott–Knot multiple comparison test ($p \leq 0.05$) and, in some cases the Dunnett test ($p \leq 0.05$) to compare the common treatments with the controls. When the ANOVA assumptions were not met, the Kruskal–Wallis test was performed, followed by the Dunn–Bonferroni test ($p \leq 0.05$). The nonlinear regression fit was used to estimate the parameters of the different epidemiological models. The relationship between vegetative, reproductive and epidemiological growth parameters was calculated using Pearson's correlation. All calculations, analyses and statistical graphs were performed with the ExpDes.pt, Treatments.ad, pliman, eppifiter and ggplot2 packages, available in R 4.1.0 software [60].

## 3. Results

### 3.1. Identification of the Fungal Isolate

Combined phylogenetic analysis of the Alt a1 and *gpd* genes grouped the isolate CML 4259 into the *Alternaria linariae* clade from the Porri section, a species previously subdivided into *A. tomaphila*, *A. cretica* and *A. subcylindrica* and currently renamed *A. linariae* [5] (Supplementary Figure S1). The Alt a1, CAL, and *gpd* datasets were efficient in discriminating the *Alternaria* species, while the ACT and ATPase datasets failed to resolve certain species. The combined analysis of the Alt a1, *gpd*, ATPase, ACT and CAL genes grouped the isolate CML 4259 together with other representative isolates of *A. linariae* (Figure 1) with the highest support (bootstrap = 95).

### 3.2. Antifungal Activity of Essential Oils against the Fungal Isolate under In Vitro Conditions

The essential oils of thyme (TEO) and lemongrass (LGEO) inhibited 100% of the mycelial growth of *A. linariae* at a concentration of 2000 ppm and were statistically superior ($p \leq 0.05$) in relation to the control treatments *Bacillus amyloliquefaciens* MBI600 (BCA MBI600), tebuconazole and azoxystrobin (Table 2). Tea tree essential oil (TTEO) showed a reduced percentage (<11%) of control in the mycelial growth inhibition (MGI) of *A. linariae*.

The lowest mean effective concentration ($EC_{50}$) required achieving half the maximum possible control effect was presented by TEO at 270 ppm, while LGEO required 1129 ppm to cause this effect. TTEO needed 1981 ppm to cause this, but its control capacity was inefficient (Supplementary Table S2).

TEO showed a higher MGI as its concentration increased, and 100% control was reached at 2000 ppm, remaining stable up to 4000 ppm, which was explained by regression analysis (Figure 2). On the other hand, LGEO showed a similar level of control as TEO, where the percentage of absolute control was achieved at a concentration of 2000 ppm,

and maintained up to 4000 ppm. Conversely, the linear regression fit indicated that TTEO reduced the control of mycelial growth, as product concentrations increased.

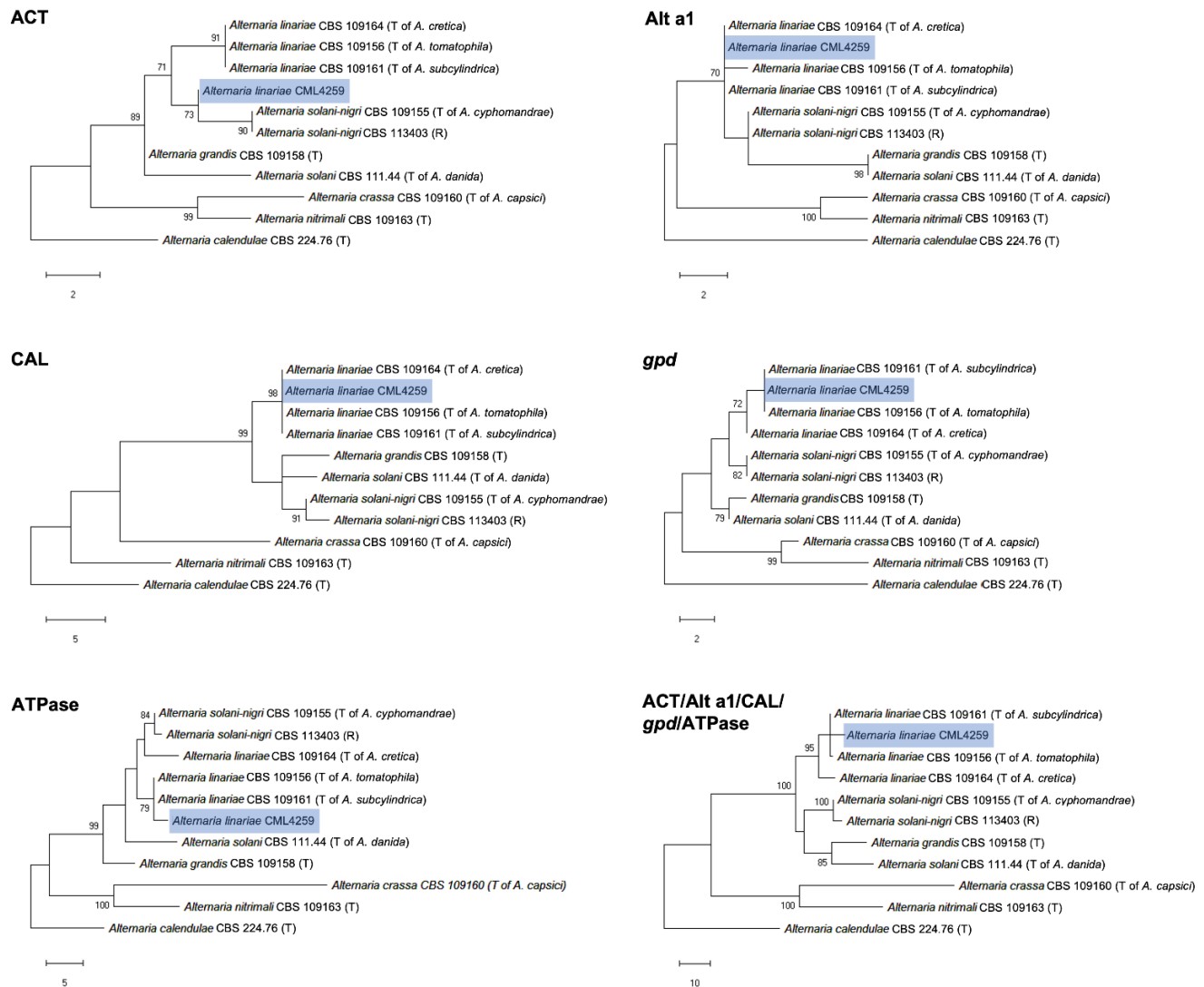

**Figure 1.** Maximum parsimony phylogenetic tree of *Alternaria* sp. The isolate from this study is indicated in bold. Bootstrap values ≥ 70% are displayed in the tree's internodes. *Alternaria calendulae* CBS 224.76 was used as an outgroup. T: ex-type strain; R: representative strain.

**Table 2.** Effect of essential oils on mycelial growth inhibition (%) of *Alternaria linariae*.

| Treatment | Concentration (ppm) | | | |
|---|---|---|---|---|
| | **500** | **1000** | **2000** | **4000** |
| Thyme essential oil | 72.26 [a,†,+] | 66.28 [a,†,*,+] | 100.00 [a,†,*,+] | 100.00 [a,†,*,+] |
| Lemongrass essential oil | 20.85 [b,†,*,+] | 32.86 [b,†,*] | 100.00 [a,†,*,+] | 100.00 [a,†,*,+] |
| Tea tree essential oil | 10.93 [c,†,*,+] | 11.14 [c,†,*,+] | 4.97 [b,†,*,+] | 1,92 [b,†,*,+] |
| *Bacillus amyloliquefaciens* MBI600 | 85.29 | | | |
| Tebuconazole | 79.83 | | | |
| Azoxystrobin | 37.04 | | | |

Means followed by the same letter in the column do not differ from each other by the Scott–Knott test ($p \leq 0.05$). Means followed by [†], [*] and [+] differ from *Bacillus amyloliquefaciens* MBI600, tebuconazole and azoxystrobin (controls), respectively, by Dunnett's test ($p \leq 0.05$).

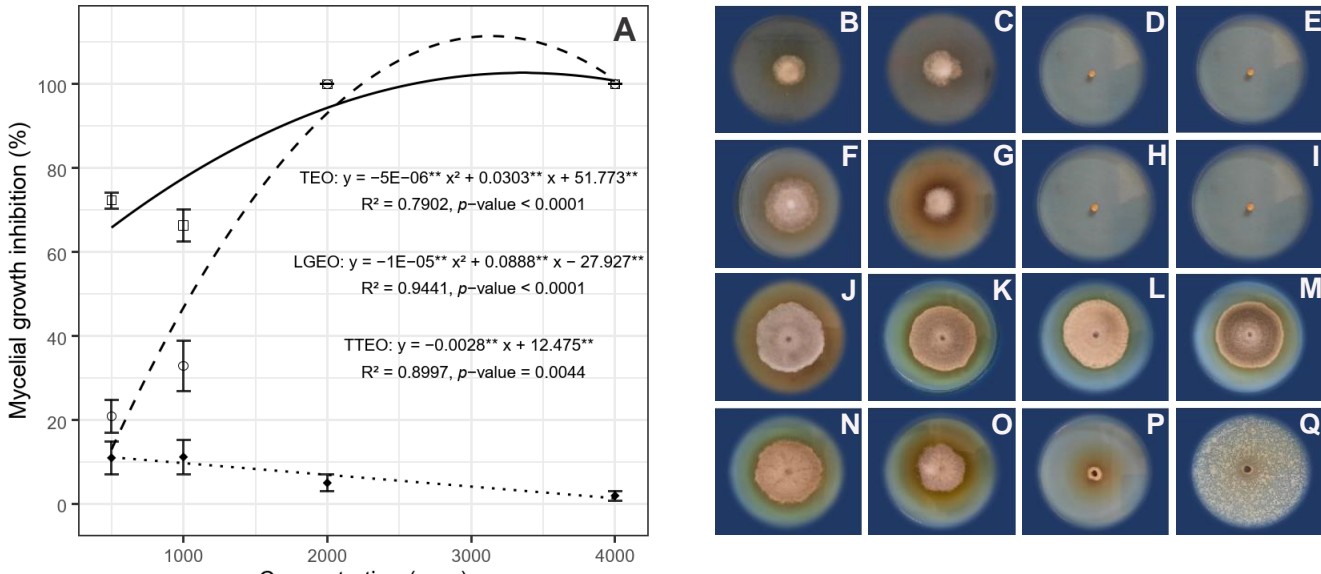

**Figure 2.** (**A**) Regression analysis between treatment concentrations and mycelial growth inhibition of *Alternaria linariae*. Treatments: thyme essential oil (TEO —□—), lemongrass essential oil (LGEO —○—) and tea tree essential oil (TTEO . . . ..♦ . . . ..). Data represent the means of values from two independent experiments (*n* = 10). ** statistical significance of the coefficients of the regression models by the F test (*p* ≤ 0.01). $R^2$, coefficient of determination. (**B**–**E**) Photographs of the mycelial growth of *A. linariae* on PDA + TEO culture medium exposed to concentrations of 500, 1000, 2000 and 4000 ppm; (**F**–**I**) PDA + LGEO at concentrations of 500, 1000, 2000 and 4000 ppm; (**J**–**M**) PDA + TTEO at concentrations of 500, 1000, 2000 and 4000 ppm, respectively; (**N**) PDA + distilled water; (**O**) PDA + Azoxystrobin; (**P**) PDA + tebuconazole; and (**Q**) PDA + *Bacillus amyloliquefaciens* MBI600.

The treatment with TEO was highly effective at controlling conidial germination of *A. linariae*, with extremely low germination percentages, fluctuating between 7.78% and 0.89% at concentrations of 500 to 4000 ppm, respectively. High levels of control were also observed in the treatment with LGEO, with germination percentages of only 15.30% to 1.11% at concentrations of 500 to 4000 ppm, respectively (Figure 3). Both EOs, at different concentrations, were statistically superior (*p* ≤ 0.05) to the control treatments BCA BMI600, tebuconazole, azoxystrobin and distilled water. On the other hand, high percentages (≥96%) of conidial germination were observed in the TTEO treatment, which indicated a reduced ability to inhibit the germination of *A. linariae* conidia.

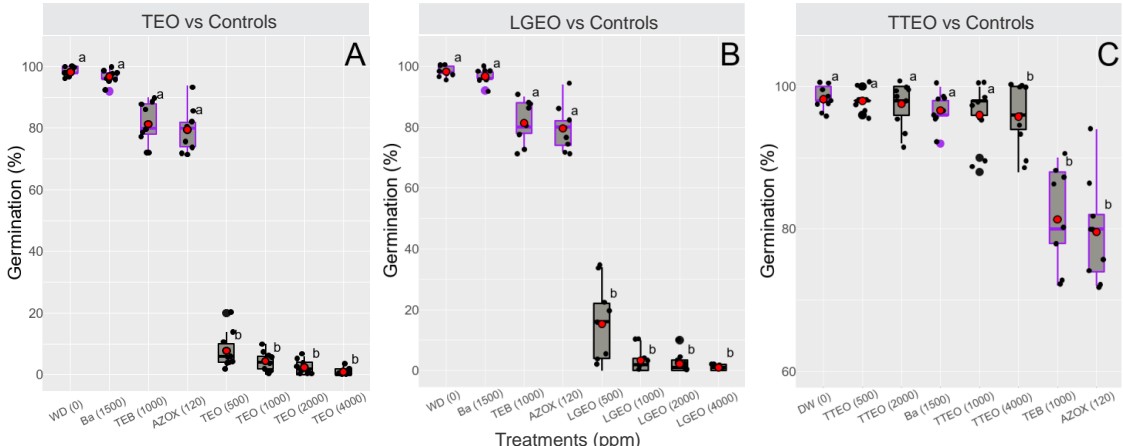

**Figure 3.** Effect of different essential oils (EOs) on the germination percentage of *Alternaria linariae* conidia. Treatments: essential oils of thyme (TEO; **A**), lemongrass (LGEO; **B**) and tea tree (TTEO; **C**).

Controls: *Bacillus amyloliquefaciens* MBI600 (Ba), tebuconazole (TEB), azoxystrobin (AZOX) and distilled water (DW). Boxes with the same letters do not differ based on the Kruskal–Wallis test ($p \leq 0.05$) with *p* value adjustment according to the Dunn–Bonferroni test ($p \leq 0.05$). Black and purple boxes represent common and control treatments, respectively. Bold points represent the distribution of values from three independent experiments ($n = 9$) and red points indicate the mean of treatments.

### 3.3. Antifungal Activity of Essential Oils against the Fungal Isolate under In Vivo Conditions

3.3.1. Infected Leaf Area

All tomato plants inoculated with *A. linariae* showed symptoms of tomato early blight (EB) at seven days after inoculation (DAI). However, the lowest percentage of infected leaf area was observed in plants treated with tebuconazole, BCA MBI600 and TEO at 6.99%, 12.70% and 20.07%, respectively, and was statistically lower ($p \leq 0.05$) than that for water-treated plants, which exhibited a 37.71% infected leaf area (Figure 4).

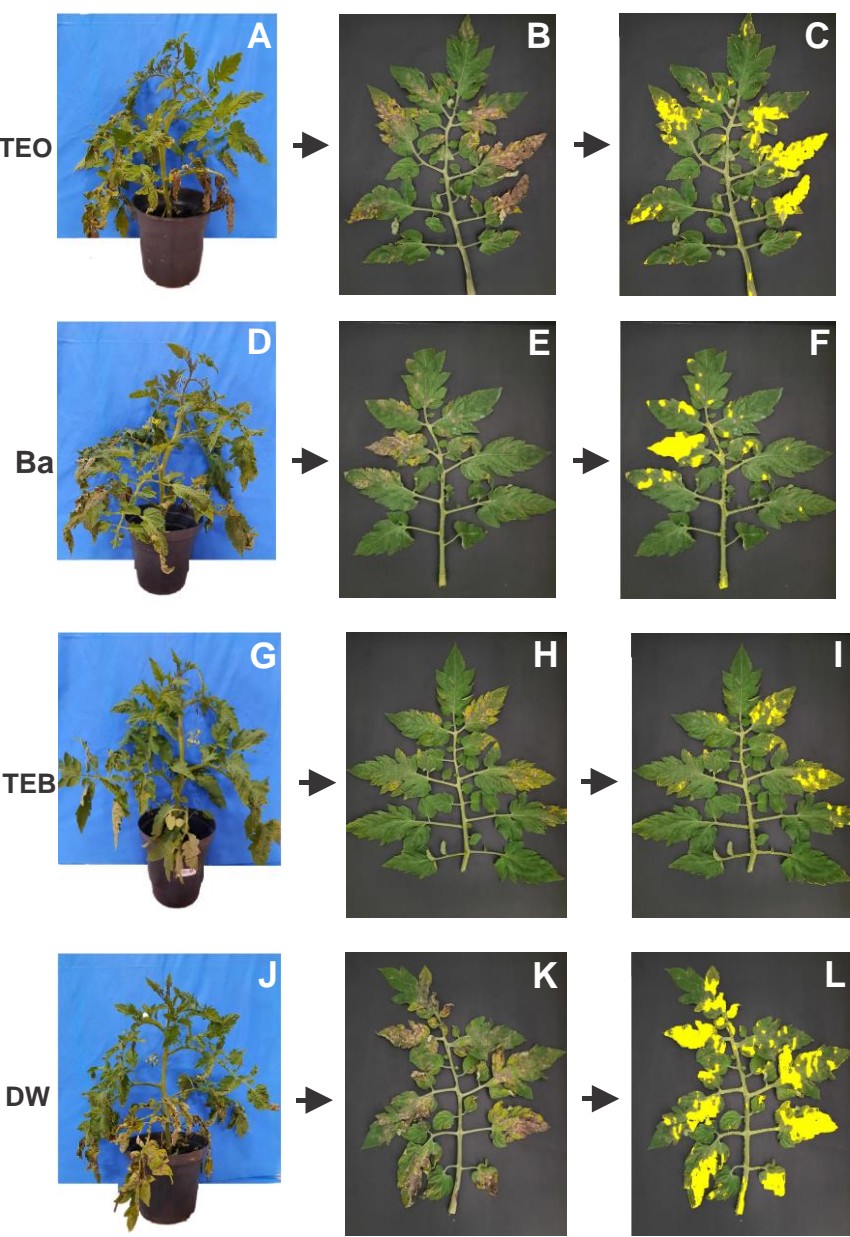

**Figure 4.** *Cont.*

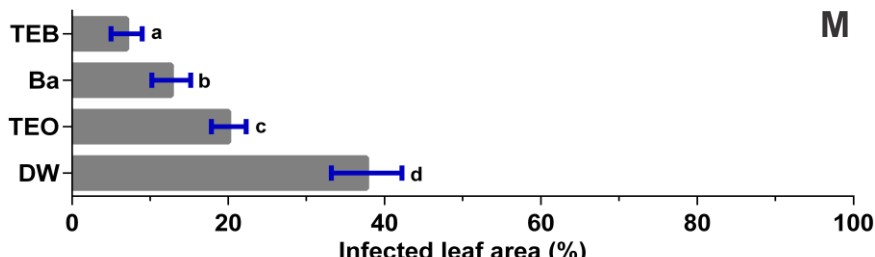

**Figure 4.** Photographs of the leaf area infected by *Alternaria linariae* in tomato plants var. Santa Clara. Tomato plants with early blight symptoms seven days after inoculation, treated with: (**A**) thyme essential oil (TEO); (**D**) biological control agent (BCA) *Bacillus amyloliquefaciens* MBI600 (Ba); (**G**) tebuconazole (TEB); and (**J**) distilled water (DW). Symptoms and infected leaf area of the third leaf (**B,E,H,K**) and estimation of the infected leaf area in plants treated with TEO, Ba, TEB, and DW (**C,F,I,L**). Bars with different letters are significantly different from each other according to the Scott–Knott test ($p \leq 0.05$) (**M**).

### 3.3.2. Disease Progress Curve, Severity and AUDPC

Among the different epidemiological models tested, the exponential model showed the best nonlinear regression fit, and more logically represented the progress of EB disease over time, with the coefficient of determination ($R^2$) of the exponential model being higher in each of the treatments evaluated (Table 3).

**Table 3.** Parameters estimated by nonlinear regression with different epidemiological models fitted to the progress of tomato early blight disease caused by *Alternaria linariae*.

| Fitting Models | Statistics | | | $r$ [d] | | $y_0$ [e] | |
| --- | --- | --- | --- | --- | --- | --- | --- |
| | CCC [a] | $R^2$ [b] | RSE [c] | Estimate | Std. Error | Estimate | Std. Error |
| **Thyme Essential Oil Treatment** | | | | | | | |
| Exponential | 0.8553 | 0.7345 | 0.0526 | 0.0223 | 0.0048 | 0.0742 | 0.0203 |
| Logistic | 0.8236 | 0.6862 | 0.0621 | 0.0268 | 0.0065 | 0.0727 | 0.0236 |
| Gompertz | 0.7717 | 0.6171 | 0.0621 | 0.0119 | 0.0033 | 0.0717 | 0.0309 |
| ***Bacillus amyloliquefaciens* MBI600 treatment** | | | | | | | |
| | CCC | $R^2$ | RSE | Estimate | Std. error | Estimate | Std. error |
| Exponential | 0.8689 | 0.7575 | 0.0427 | 0.0221 | 0.0045 | 0.0649 | 0.0166 |
| Logistic | 0.8451 | 0.7202 | 0.0456 | 0.0258 | 0.0059 | 0.0636 | 0.0188 |
| Gompertz | 0.8015 | 0.6588 | 0.0499 | 0.0111 | 0.0028 | 0.0615 | 0.0243 |
| **Tebuconazole treatment** | | | | | | | |
| | CCC | $R^2$ | RSE | Estimate | Std. error | Estimate | Std. error |
| Exponential | 0.8867 | 0.7886 | 0.0272 | 0.0174 | 0.0032 | 0.0688 | 0.0120 |
| Logistic | 0.8729 | 0.7665 | 0.0285 | 0.0200 | 0.0039 | 0.0679 | 0.0132 |
| Gompertz | 0.8438 | 0.7230 | 0.0309 | 0.0084 | 0.0018 | 0.0660 | 0.0162 |
| **Distilled water treatment** | | | | | | | |
| | CCC | $R^2$ | RSE | Estimate | Std. error | Estimate | Std. error |
| Exponential | 0.9449 | 0.8932 | 0.0496 | 0.0174 | 0.0022 | 0.1839 | 0.0220 |
| Logistic | 0.9076 | 0.8292 | 0.0623 | 0.0273 | 0.0047 | 0.1723 | 0.0318 |
| Gompertz | 0.8813 | 0.7888 | 0.0692 | 0.0166 | 0.0032 | 0.1626 | 0.0412 |

[a] CCC, Lin's concordance correlation coefficient; [b] $R^2$, coefficient of determination; [c] RSE, residual standard error; [d] r, infection rate; [e] $y_0$, initial inoculum.

From seven DAI, tomato plants exhibited different levels of disease (EB) severity. Plants treated with TEO controlled the expression of symptoms and disease severity at levels from 20% up to 50 DAI, similar to controls BCA MBI600 and tebuconazole, but were significantly different from plants treated with water, which at that time of evaluation showed approximately 50% disease severity (Figure 5; Supplementary Figure S2).

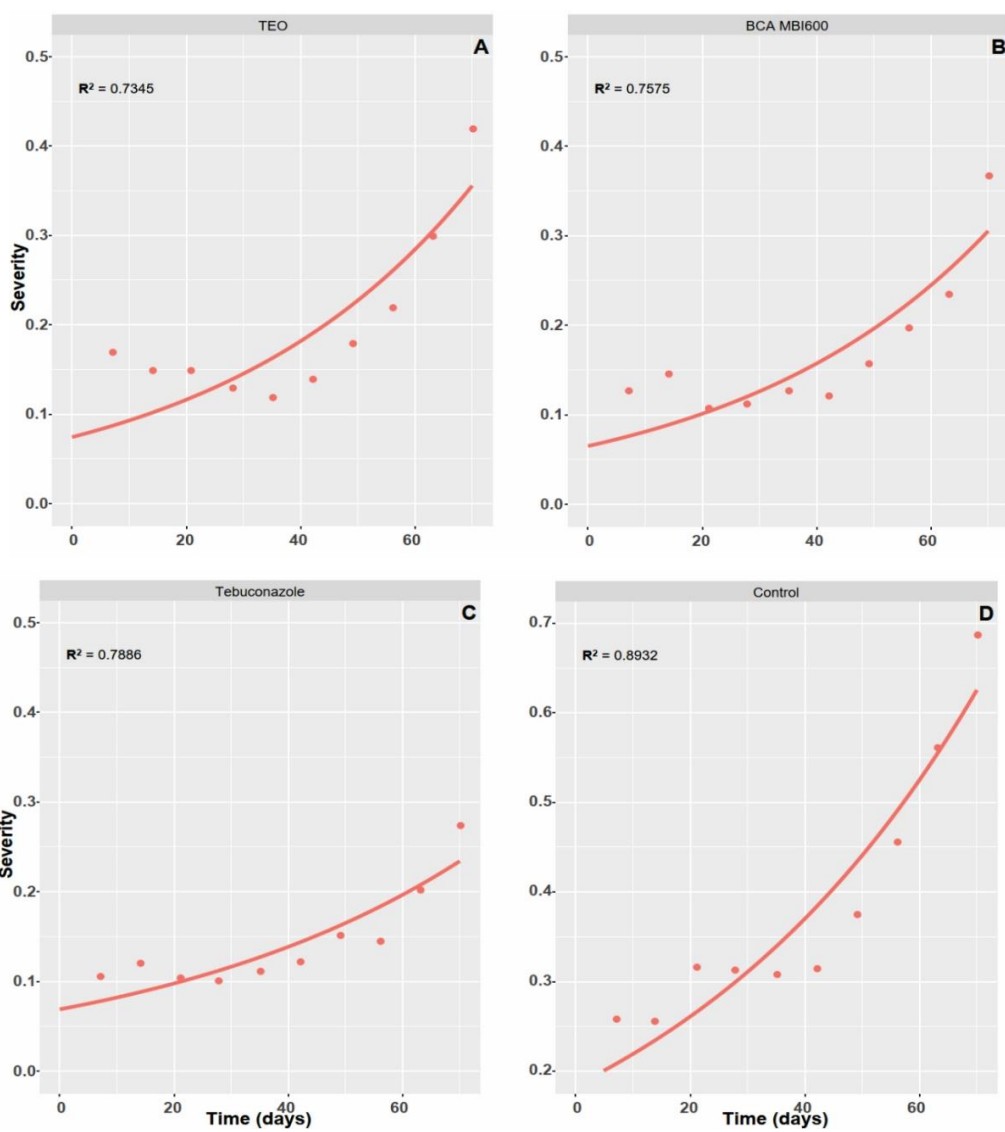

**Figure 5.** Exponential model fitted curves (lines) to disease progress of tomato early blight under different treatments: (**A**) thyme essential oil (TEO), (**B**) *Bacillus amyloliquefaciens* MBI600 (BCA MBI600), (**C**) tebuconazole, and (**D**) control (distilled water). Values (points) represent the mean (*n* = 12) of the disease index evaluated weekly. $R^2$, coefficient of determination.

From 50 DAI, there was a significant increase in disease severity, and at 70 DAI, treatments showed statistically significant differences ($p \leq 0.05$). Treatment with TEO reduced EB severity by 39% when compared to water-treated plants, while BCA MBI600 and tebuconazole reduced disease severity by 46% and 60%, respectively. Therefore, the final disease severity was 27.50%, 36.74%, and 42.00% in the treatments with tebuconazole, BCA MBI600 and TEO, respectively, and was significantly lower ($p \leq 0.05$) than that in the water treatment, with a severity of 68.80% (Figure 6). The cumulative disease intensity over time was expressed as AUDPC, and the highest values were observed in water-treated plants, with an AUDPC of 2366. In turn, the TEO treatment presented an AUDPC of 1176, which was significantly lower ($p \leq 0.05$) than that of the water treatment. However,

tebuconazole and BCA MBI600 exhibited the lowest AUDPC values at 877 and 1020, respectively, which resulted in lower disease intensity at the end of the crop cycle (Figure 6).

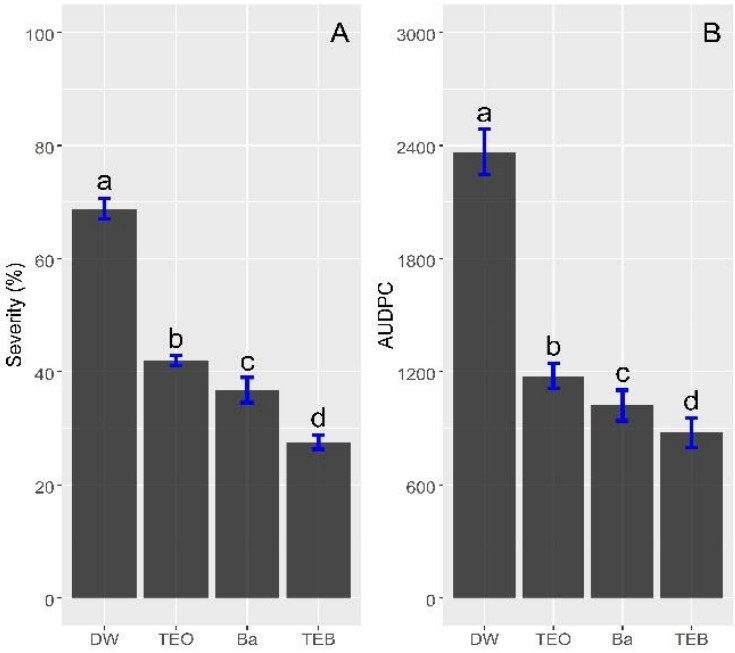

**Figure 6.** (**A**) Final disease severity (disease index) of tomato early blight caused by *Alternaria linariae* at 70 days after inoculation. Bars followed by different letters differ from each other by the Scott–Knott test ($p \leq 0.05$). (**B**) Area under the disease progress curve (AUDPC). Mean AUDPC values are the result of the cumulative disease index. Treatments: DW: distilled water; TEO: thyme essential oil; Ba: *Bacillus amyloliquefaciens* MBI600; and TEB: tebuconazole. Bars with different letters are significantly different from each other according to the Scott–Knott test ($p \leq 0.05$).

### 3.4. Vegetative Growth and Yield Parameters

There were statistically significant differences in vegetative growth parameters ($p \leq 0.05$). The plant height in the treatments with TEO, BCA MBI600 and tebuconazole was 128.78, 135.31 and 139.23 cm, respectively, which was significantly higher than that in water treatment, which was 120.44 cm tall. The stem diameter was reduced by an average of 13.22% in the water treatment compared to the rest of the treatments. Plants treated with only water showed a reduced number of leaves by 39.30%, 46.79% and 49.41% in relation to treatments with TEO, BCA MBI600 and tebuconazole, respectively. The fresh weight of plants in the water treatment decreased by 29.60%, 39.60% and 51.32% compared to the TEO, BCA MBI600 and tebuconazole treatments, respectively. However, the dry weight in water-treated plants decreased by 29.69%, 37.98% and 47.13% in relation to the TEO, BCA MBI600 and tebuconazole treatments, respectively (Table 4).

The number of fruits was not affected by the treatments. The average fruit weight in the water treatment was 20.00%, 23.10% and 23.85% lower than that in the TEO, BCA MBI600 and tebuconazole treatments, respectively. However, the productivity/plant in the water treatment decreased by 24.79%, 31.78% and 31.00% when compared to the TEO, BCA MBI600 and tebuconazole treatments, respectively. The TEO treatment showed no significant differences ($p \leq 0.05$) in yield/plant compared to the biological and chemical controls BCA MBI600 and tebuconazole, respectively (Table 5).

**Table 4.** Effect of the application of thyme essential oil on the vegetative growth parameters of tomato plants infected by *Alternaria linariae*.

| Treatments | Vegetative Growth Parameters | | | | |
|---|---|---|---|---|---|
| | Plant Height (cm) | Stem Diameter (mm) | No. Leaves | Plant Fresh Weight (g) | Plant Dry Weight (g) |
| Thyme essential oil | 128.78 ± 4.99 [b] | 16.17 ± 0.66 [a] | 11.50 ± 0.52 [c] | 466.75 ± 120.07 [c] | 97.88 ± 28.95 [b] |
| *B. amyloliquefaciens* MBI600 | 139.23 ± 3.56 [a] | 16.00 ± 0.87 [a] | 12.83 ± 0.83 [b] | 543.92 ± 93.83 [b] | 110.95 ± 29.84 [b] |
| Tebuconazole | 135.31 ± 4.37 [a] | 16.17 ± 0.78 [a] | 13.50 ± 0.52 [a] | 675.00 ± 128.78 [a] | 130.16 ± 28.98 [a] |
| Distilled water | 120.44 ± 6.95 [c] | 13.98 ± 0.82 [b] | 6.83 ± 0.94 [d] | 328.58 ± 64.92 [d] | 68.82 ± 14.30 [c] |
| CV (%) * | 3.91 | 6.98 | 6.52 | 20.83 | 25.82 |

Values show the means ± standard deviation (*n* = 12). Data within the same column followed by different letters are significantly different by the Scott–Knott (*p* ≤ 0.05). * CV, Pearson's coefficient of variation.

**Table 5.** Effect of the application of thyme essential oil on the yield parameters of tomato plants infected by *Alternaria linariae*.

| Treatments | Reproductive Parameters | | |
|---|---|---|---|
| | Average Fruit Weight (g) | No. Fruits/Plants | Average Plant Yield (Kg) |
| Thyme essential oil | 125.91 ± 9.56 [a] | 18.67 ± 2.67 [a] | 2.34 ± 0.32 [a] |
| *B. amyloliquefaciens* MBI600 | 131.11 ± 5.48 [a] | 19.67 ± 1.72 [a] | 2.58 ± 0.27 [a] |
| Tebuconazole | 132.39 ± 5.92 [a] | 19.33 ± 2.71 [a] | 2.55 ± 0.36 [a] |
| Distilled water | 100.82 ± 7.56 [b] | 17.50 ± 1.45 [a] | 1.76 ± 0.13 [b] |
| CV (%) * | 5.96 | 11.6 | 12.3 |

Values show the means ± standard deviation (*n* = 12). Data within the same column followed by different letters are significantly different by the Scott–Knott (*p* ≤ 0.05). * CV Pearson's coefficient of variation.

*3.5. Pearson's Correlation*

A significant negative correlation was observed between the epidemiological parameters of disease severity and AUDPC and the parameters of vegetative growth and productivity. Disease severity and AUDPC negatively affected stem diameter, plant height, number of leaves, and wet and dry plant weight. The average fruit weight and yield/plant decreased as disease severity and AUDPC increased. However, the average number of fruits was not related to the epidemiological variables (Figure 7).

*3.6. Defense-Related Enzyme Activity*

At 144 h after inoculation (HAI), TEO treatment increased the enzymatic activity of peroxidase (POX) by 43.30% in relation to inoculated plants, and was statistically equal to treatment with BCA MBI600 (*p* ≤ 0.05) (Figure 8A). Plants sprayed with tebuconazole and uninoculated exhibited low POX activity.

BCA MBI600 application increased polyphenol oxidase (PPO) activity at 144 HAI by 37.15 and 43.33% compared to TEO treatment and inoculated plants (Figure 8B). At 192 HAI, the highest activity of this enzyme was observed in plants treated with TEO and BCA MBI600, although this activity was not significantly different from that of the inoculated plants.

Phenylalanine ammonia-lyase (PAL) activity showed a constant increase over time; however, only at 96 HAI did plants treated with TEO and BCA MBI600 show an increase in PAL activity by 30.30% and 27.28%, respectively, which was significantly different from the inoculated plants (*p* ≤ 0.05) (Figure 8C). At 144 and 192 HAI, the plants treated with TEO did not show significant increases in relation to the control treatments.

BCA MBI600 treatment showed an increase of 65.20% in β-1,3-glucanase (GLU) activity compared to plants inoculated at 48 HAI. However, at 144 and 192 HAI, TEO applications improved GLU activity by 25.50% and 20.70%, respectively, which was significantly higher (*p* ≤ 0.05) than that for inoculated plants (Figure 8D).

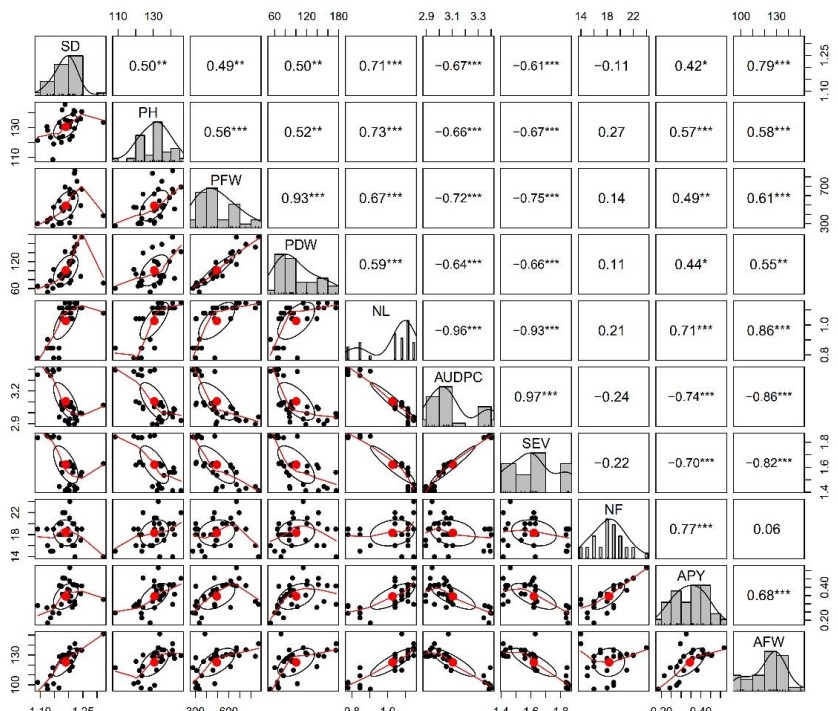

**Figure 7.** Pearson's correlation between variables associated with vegetative growth, yield and epidemiological parameters (top right panel). Histograms (perpendicular centerline). Correlation fit line and data distribution (bottom left panel). Variables: SD, stem diameter; PH, plant height; PFW, plant fresh weight; PDW, plant dry weight; NL, no. of leaves; AUDPC, area under the disease progress curve; SEV, disease severity; NF, no. of fruits; APY, average plant yield; and AFW, average fruit weight. * Indicates $p < 0.05$, ** indicates $p < 0.01$, *** indicates $p < 0.001$.

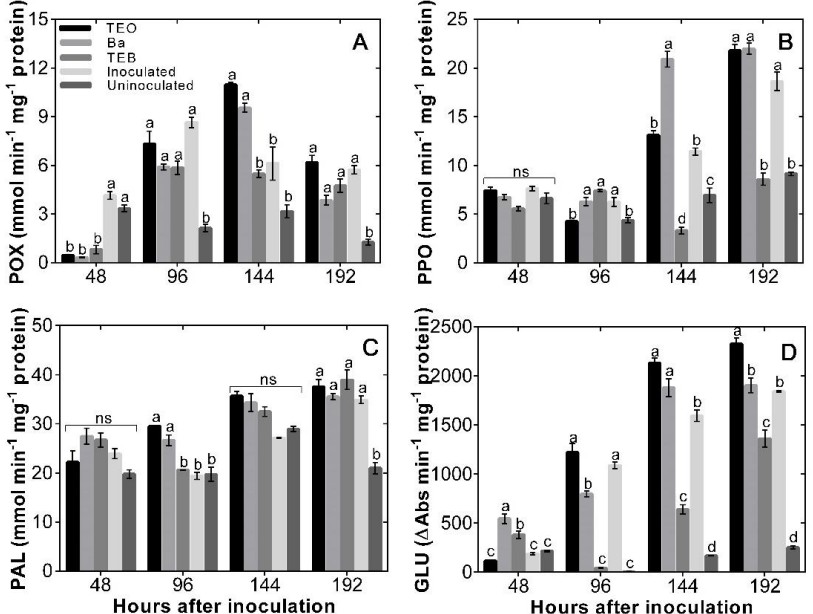

**Figure 8.** Defense-related enzyme activity in tomato plants var. Santa Clara infected by *Alternaria linariae*. (**A**) Peroxidase (POX), (**B**) polyphenol oxidase (PPO), (**C**) phenylalanine ammonia-lyase (PAL) and (**D**) β-1,3-glucanase (GLU). Treatments: thyme essential oil (TEO), *Bacillus amyloliquefaciens* MBI600 (Ba), tebuconazole (TEB); treatments were performed on inoculated and uninoculated plants. Bars with the same letters are not significantly different from each other by the Scott–Knott test ($p \leq 0.05$). Data represent the average absorbance at different collection times, and deviation bars indicate the standard error of the mean.

### 3.7. Histochemical Analysis of Foliar Tissues Infected or Not by Alternaria linariae

TEO and BCA MBI600 treatments increased the accumulation of phenolic compounds in the leaf tissue of tomato plants infected with EB. This accumulation occurred at 48 HAI and was maintained up to 192 HAI (Figure 9). Inoculated plants showed minimal accumulation of these compounds, while uninoculated plants showed no histological changes associated with the accumulation of phenolic compounds. On the other hand, callose accumulation increased in tomato plants treated with TEO and BCA MBI600. The fluorescence intensity observed in the tissues was higher in relation to inoculated and uninoculated plants (Figure 9). These histochemical responses associated with TEO and BCA MBI600 applications were maintained over time, up to 192 HAI.

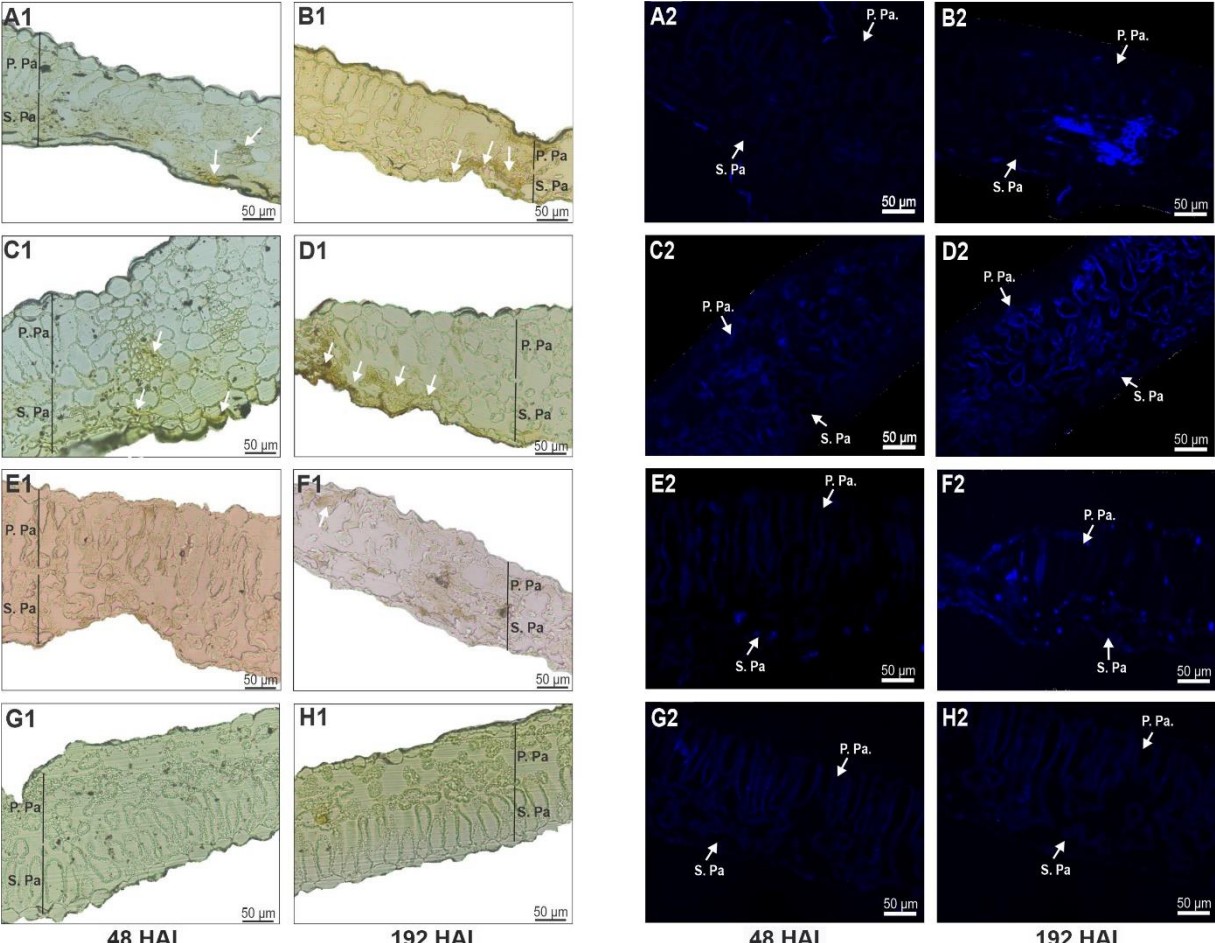

**Figure 9.** Photomicrographs (**A1**–**H1**) and epifluorescence images (**A2**–**H2**) showing phenolic compound accumulation and callose deposition, respectively, in leaf tissues of tomato plants infected by *Alternaria linariae* at 48 and 192 h after inoculation (HAI). Micrographs of transverse leaf sections (8 μm width) previously embedded in historesin. Treatments: thyme essential oil (**A**,**B**); *Bacillus amyloliquefaciens* MBI600 (**C**,**D**); inoculated plants (**E**,**F**); and uninoculated plants (**G**,**H**). Arrows (left panel) indicate accumulation of phenolic compounds (with a color between brown and dark yellow) after staining with ferric chloride. P. Pa. and S. Pa. indicate the palisade and spongy parenchyma tissues, respectively. Leaf tissue regions with high fluorescence (right panel) intensity indicate greater accumulation of callose after staining with aniline blue (0.005%) under observation with a Zeiss 49 blue filter (excitation = 365 nm; emission 445/50 (420–470 nm).

## 4. Discussion

Essential oils (EOs) have been proposed as alternative and eco-friendly products to control phytopathogens [21]. The numerous components of EOs have multiple mechanisms

of action that alter several cellular biological processes [34]. Our findings showed that thyme (TEO) and lemongrass (LGEO) were highly efficient at inhibiting the mycelial growth of *Alternaria linariae* under in vitro conditions. This effect was previously associated with the activity of thymol and citral, the main components of TEO and LGEO, respectively [29,61], constituting compounds with high antifungal activity [62,63].

In this study, TEO showed high fungicidal activity, and effectively controlled the mycelial growth and conidial germination of *A. linariae*. Previous studies reported that TEO inhibited the mycelial growth of *A. solani* [64], and conidia germination of *A. alternata* [65]. Mycelial grown inhibition (MGI) improved as EO concentrations increased [65], which is in agreement with our findings, where mycelial growth was negatively affected by increasing TEO concentrations. The antifungal activity of TEO was associated with altered fatty acid (ergosterol) metabolism and increased oxidative stress by the accumulation of reactive oxygen species (ROS) [61]. Similar to TEO, LGEO effectively reduced mycelial growth and conidial germination of *A. linariae*. LGEO has been reported to strongly inhibit the mycelial growth of *A. solani*, *A. alternata*, *A. tenuissina* and *Fusarium verticillioides* [29,66,67]. The biological activity of LGEO affects different cellular structures and functions, such as the alteration of mycelium morphology, increased cell permeability [68] and decreased lipid content [69]; therefore, LGEO is considered a multisite antifungal agent [63]. On the other hand, although tea tree EO (TTEO) has been reported to control several phytopathogens, such as *Botrytis cinerea* [70], we did not observe antifungal effects on the inhibition of mycelial growth or on the germination of *A. linariae* conidia.

TEO foliar applications reduced the symptoms and severity of tomato early blight (EB) disease. Plants treated with TEO exhibited better health compared to the water treatment, which showed a significant reduction in leaf area. Previous studies showed that TEO presented postharvest phytopathogen control such as *B. cinerea* and *Colletotrichum musae* [71,72]. However, under in planta conditions, it reduced the severity of gray mold (*B. cinerea*) and tomato wilt caused by *F. oxysporum* f. sp. *radicis-lycopersici* [37]. Ultrastructural analysis showed that the antifungal activity of TEO phenolic substances limits the growth and reduces the fungal viability of *A. alternata*, delaying the penetration of tangerine fruit tissues by causing interference in the membrane and cell wall [73]. The antifungal properties previously reported for TEO could have explained the reduction in the severity of EB; therefore, foliar applications of TEO could have important benefits in the management of this disease and become a convenient tool in preventing the development of fungicide resistance by populations of *Alternaria* spp., which are considered etiological agents of EB.

In this study we observed the positive effect of TEO foliar applications on vegetative growth and tomato yield parameters. A similar effect was observed in tomato plants infected by *Phytophthora infestans* and *Alternaria solani* and treated with essential oils of ginger, garlic, and Mexican marigold, where plant growth and yields were significantly boosted [13].

Induced systemic resistance (ISR) is an active resistance mechanism that is dependent on physicochemical barriers elicited by agents of biotic or abiotic origin [74]. Thus, ISR has been elicited by the application of inducers (elictors) of biological origin, including plant growth-promoting rhizobacteria (PGPR) and, in fewer reports, plant EOs [20,75]. Among these mechanisms that contribute to the improvement of plant defense against the attack of phytopathogens, there is the increase in the activity of defense-related enzymes [76].

Our findings showed that TEO application elicited an increase in the activity of peroxidase (POX), polyphenol oxidase (PPO) and β-1,3-glucanase (GLU), and to a minor extent phenylalanine ammonia lyase (PAL). Previously, TEO was reported to activate defense-related enzymes and genes in the control of various phytopathogens [37,71,77]. TEO induced an increase in POX activity in tomato plants, which was related to a reduced severity of gray mold disease and tomato wilt caused by *B. cinerea* and *F. oxysporum* f. sp. *lycopersici*, respectively [37]. Similarly, TEO and a chitosan coating treatment increased the expression of POX, PAL, GLU and chitinase in the control of avocado (*Persea americana*) anthrac-

nose [77]. At postharvest, TEO triggered the defense response in apple (*Malus domestica*) fruits, with increased transcriptional levels of PR-8 gene mRNA, which encodes chitinase, and contributed to the reduction in gray mold caused by *B. cinerea* [71]. On the other hand, PGPRs such as *Bacillus amyloliquefaciens* are capable of eliciting ISR through defense-related enzymes and genes such as *PR-1a*, *LOX*, *ERF1* and *PDF1.2*, which are activated by the dependent signaling pathways of salicylic acid (SA) and jasmonic acid/ethylene (AJ/ET) [78,79]. In some cases, these signaling pathways have been reported to be antagonists [79,80], although there are also reports where both signaling pathways (SA and JA/ET) act in synergism [16,81]. In this work, TEO sometimes equaled or even improved the activity of some of the enzymes studied here, such as POX, PPO and GLU, compared to the control treatment *B. amyloliquefaciens* MBI600.

The outstanding POX activity in TEO-treated plants could have regulated ROS levels at relatively lower toxic concentrations in *A. linariae*-infected plants. Excess $H_2O_2$ can lead to an increase in POX activity [82]. Thus, it can act as a cosubstrate in the oxidation of phenolic compounds while consuming $H_2O_2$ [83,84]. In turn, PPO commonly interacts with POX to promote ROS scavenging [85]. In our study, both enzymes showed similar activity over time, indicating a possible synergism between these two enzymes. PPO is also important in the oxidation of phenolic compounds, forming quinones, which can crosslink or alkylate proteins [86]. This o-quinone–protein complex, formed as a consequence of cell damage, reduces the cellular nutritional value, decreasing the rate of advance to the pathogen in the tissues [87]. We observed that PAL activity, although not significant at some evaluation times, increased over time and was often larger in the TEO treatment. PAL, through the deamination of phenylalanine, produces the formation of trans-cinnamic acid, a precursor of salicylic acid (SA) biosynthesis [88]. Therefore, PAL activity improves SA synthesis, and thus, the induction of systemic resistance [89]. Furthermore, PAL is responsible for activating the phenylpropanoid pathway, which is important in phenol synthesis [90]. On the other hand, TEO applications induced higher GLU activity than BCA MBI600 and the rest of the treatments. The hydrolytic activity of GLU plays an important role in the degradation of β-1,3/1,6-glucan bonds in the walls of plant parasite fungi, inhibiting their growth and releasing large amounts of elicitors, which induce the synthesis of phytoalexins and phenolic compounds [91,92] Such facts could have contributed to the improvement of resistance and control of EB in tomato plants infected by *A. linariae*.

Phenolic compound accumulation and callose deposition in *A. linariae*-infected tissues increased due to TEO treatment. This effect was observed at 48 h after inoculation (HAI) and was maintained over time, up to 192 HAI, similar to the biological control BCA MBI600, and was higher than the rest of the treatments. Therefore, phenolic compound accumulation and callose deposition due to the effect of the TEO treatment improved the formation of structural barriers, which could have limited the advance of the pathogen in the sites of infection. Limited information is available on EOs inducing resistance, rather than PGPR, where this type of effect is well known [89,93,94]. Often, callose deposition initially occurs below the sites of fungal appressorium formation [75]. However, callose deposition can be observed at sites distal to the leaf epidermis area as colonization progresses [95]. This fact agrees with our findings, initially, callose deposition was observed in tissues close to the epidermis, progressing to the mesophyll cells, and finally, at 192 HAI, it was dispersed throughout the tissue in the cross section of the leaf blade. Previous studies have shown that the accumulation of phenols by the application of EOs had a significant effect in improving antioxidant activity in blueberry plants [96]. Furthermore, it was observed that TEO increased the content of phenolic compounds in tomato plants infected by gray mold (*B. cinerea*) and tomato wilt (*F. oxysporum* f. sp. *lycopersici*), and this was associated with an increase in ROS production mediated by POX activity [36]. This result indicates that the accumulation of phenolic compounds in plants results from qualitative changes in respiration processes, which act as electron carriers in oxidation reactions catalyzed by polyphenol oxidases [13]. Our results indicate that the accumulation of phenolic compounds in tomato leaf tissues infected with *A. linariae* and treated with TEO

was probably related to the high activity of POX and PPO. The defensive properties of phenolic compounds could play an important role in reducing inoculum production in the field, as they have been reported to inhibit the sporulation, spore germination and germinal hyphal growth of phytopathogenic fungi [14].

Despite studies such as the one developed by Mugao et al. [13] evaluated the effect of essential oils on vegetative growth, reproductive and epidemiological parameters in tomato plants infected by early blight caused by *Alternaria solani*, biochemical aspects and cellular anatomical responses induced by the application of some essential oils (ginger, garlic, and Mexican marigold) were not analyzed, thus, limited information of essential oils on the activation of plant defense systems was presented, different to what was explored in our study.

Finally, in this work, TEO showed high antifungal activity and effected the induction of resistance, with the activation of defense-related enzymes and other biochemical changes associated with the accumulation of phenolic compounds and callose in tomato plants infected by *A. linariae* (Supplementary Figure S3). This qualifies TEO as a product with multiple effects and high potential for EB control. To our knowledge, there are few reports available on foliar applications of TEO in reducing the severity of EB in tomato plants; therefore, this study is one of the first to show this potential. The broad expectations that TEO can be inserted into rotation programs with synthetic chemical fungicides could contribute greatly to the fight against resistance to fungicides currently used for EB control, such as strobilurins, boscalides and triazoles.

**Supplementary Materials:** The following supporting information can be downloaded at: https://www.mdpi.com/article/10.3390/horticulturae8100919/s1, Table S1: *Alternaria* spp. strains used in phylogenetic analysis and their GenBank accession numbers; Figure S1: Phylogenetic analysis using maximum parsimony of *Alternaria* spp. from the Porri section based on the Alt a1 and *gdp* gene sequences; Table S2. Effective concentration (50%) ($EC_{50}$) of essential oils on *Alternaria linariae* under in vitro conditions; Figure S2. Photographs of tomato plants var. Santa Clara with symptoms of early blight at 50 days after inoculation with *Alternaria linariae*. Figure S3. Thyme essential oil (TEO) exhibits high antifungal activity in in vitro and in vivo conditions against *Alternaria linariae*.

**Author Contributions:** Conceptualization, L.A.S.-R.; methodology, L.A.S.-R., P.R.D.S., D.G.T., D.M.D.S.B., T.A.M. and S.I.M.; software, L.A.S.-R. and D.G.T.; validation, E.A.; formal analysis, E.A.; investigation, L.A.S.-R., P.R.D.S. and D.G.T.; resources, E.A.; writing—original draft preparation, L.A.S.-R.; writing—review and editing, L.A.S.-R.; supervision, E.A.; project administration, E.A.; funding acquisition, E.A. All authors have read and agreed to the published version of the manuscript.

**Funding:** This research was funded by CAPES and CNPq 306133/2021 and CNPq PDJ-scholarship for coauthor: S.I.M.

**Data Availability Statement:** Not applicable.

**Acknowledgments:** We are grateful to Embrapa Hortaliças for providing the fungal isolate.

**Conflicts of Interest:** The authors declare no conflict of interest.

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
