# Peer review of "Thyme Essential Oil Reduces Disease Severity and Induces Resistance against Alternaria linariae in Tomato Plants"

_horticulturae, doi:10.3390/horticulturae8100919_

Round 1

Reviewer 1 Report

The manuscript entitled “Thyme essential oil reduces disease severity and induces resistance against Alternaria linariae in tomato plants”

by Luis Alberto Saltos-Rezabala, Patrícia Ricardino Da Silveira, Dérica Gonçalves Tavares, Silvino Intra Moreira, Thiago Alves Magalhães, Deila Magna Dos Santos Botelho, Eduardo Alves discusses how the application of Essential Oils (EO) on tomato plant reduces disease severity and improves resistance against Alternaria linariae, in Lycopersicum esculentum.

Reviewer’s comments:

As phytopathogens continue to severely hamper crop production and agricultural yields, biological resources and products define attractive approaches to plant disease management.

While studies have shown the importance of essential oils as antimicrobials, Mugao et al. (2021) discussed the beneficial effects of essential oils as biocontrol agents of early and late blight disease of tomato (Lydia G. Mugao, Bernard M. Gichimu, Phyllis W. Muturi, Ezekiel K. Njoroge, "Essential Oils as Biocontrol Agents of Early and Late Blight Diseases of Tomato under Greenhouse Conditions", International Journal of Agronomy, vol. 2021, Article ID 5719091, 10 pages, 2021. https://doi.org/10.1155/2021/5719091),

how this study is crucial to disease management in tomato plants, needs to be emphasized, compared, and contrasted with similar studies in this field.

However, the authors have provided key insights into the essential oil applications and disease management in tomato, application of EOs of thyme (Thymus vulgaris), lemongrass (Cymbopogon citratus), and tea tree (Melaleuca alternifolia), Histochemical analysis of Leaf Tissues, defense-related enzyme activity, plant growth and yield etc. the study discuss interesting findings and provides a platform for further exploration of bio-based products in plant disease management.

Considering the major loss in crop production due to phytopathogens, this feasible experimental study defines key prospects in combating the harmful effects of phytopathogens. I recommend the publication of the paper.

Author Response

We agree with the importance and relevance of the study mentioned by the Reviewer for the area of our research: Mugao, L.G.; Gichimu, B.M.; Muturi, P.W.; Njoroge, E.K. Essential Oils as Biocontrol Agents of Early and Late Blight Diseases of Tomato under Greenhouse Conditions. Int. J. Agron. 2021, 2021, 1–10, doi:10.1155/2021/5719091). This article has been added to the Discussion section of our article as requested by the Reviewer.

Reviewer 2 Report

In the manuscript  entitled “ Thyme essential oil reduces disease severity and induces resistance against Alternaria linariae in tomato plants”, authors describe the in vitro and in vivo antifungal activity of thyme essential oil against Alternaria linariae, a pathogen responsible for tomato early blight (EB).

Accepted with minor revisions.

I have some general  comments:

1)Which is the importance of Alternaria linariae in EB? Please introduce it.

2)Why did authors continue the in vivo study only with TEO?

Other comments:

1)      Please revise line 371 (it)

2)      Figure 8: ns?

3)      810: Alternaria solani, this is the first time this species is named

Author Response

1)Which is the importance of Alternaria linariae in EB? Please introduce it.

Answer: The importance of Alternaria linareae in EB was introduced in the Introduction session, as requested by Reviewer 2.

2)Why did authors continue the in vivo study only with TEO?

Answer: The work was continued in vivo with TEO, because this essential oil was the one that showed the best responses in the in vitro tests, with the most promising results for the in vivo tests.

Other comments:
1) Please revise line 371 (it)

Answer: The correction was made in the text.

2) Figure 8: ns?

Answer: ns means not significant. The meaning has been added in the legend of Figure 8.

3) 810: Alternaria solani, this is the first time this species is named.

Answer: Alternaria solani was first named in the Introduction session (line 34).

Reviewer 3 Report

Dear authors, this is an excellent study that covered all important aspects and analyses relevant to our understanding of EO influence on disease development.

All parts are very well written thus I have no further comments on how to improve the manuscript.

The paper title is slightly not in alignment with the paper's goals. I suggest:

'Essential oil influence on disease severity and induction of resistance against Alternaria linaria in tomato plants' since thyme was not the only EO investigated herein.

LIne 371 delete the unnecessary word 'it'

Thank you.

Author Response

The paper title is slightly not in alignment with the paper's goals. I suggest:

'Essential oil influence on disease severity and induction of resistance against Alternaria linaria in tomato plants' since thyme was not the only EO investigated herein.

Answer: We agree with a suggestion to change the title by the Reviewer. The title has been changed in the article as requested by the Reviewer.

Line 371 delete the unnecessary word 'it'

Answer: The correction was made in the text.